# Inoculation with *Oidiodendron maius* BP Improves Nitrogen Absorption from Fertilizer and Growth of *Vaccinium corymbosum* during the Early Nursery Stage

**DOI:** 10.3390/plants12040792

**Published:** 2023-02-10

**Authors:** María A. Pescie, Marcela Montecchia, Raul S. Lavado, Viviana M. Chiocchio

**Affiliations:** 1IIPAAS (Instituto de Investigación en Producción Agropecuaria, Ambiente y Salud), Facultad de Ciencias Agrarias, Universidad Nacional de Lomas de Zamora, Ruta 4, Km. 2, Llavallol, Lomas de Zamora 1836, Provincia de Buenos Aires, Argentina; 2INBA (CONICET-UBA), Facultad de Agronomía, Universidad de Buenos Aires, Av. San Martín 4453, Buenos Aires C1417DSE, Argentina

**Keywords:** blueberry plantlets, nitrogen absorption, tagged nitrogen, fungal strains

## Abstract

Blueberry roots are inefficient in taking up water and nutrients, a fact partially related to their scarcity of root hairs, but they improve nutrient uptake by associating with ericoid mycorrhizal and endophytic fungi. However, the benefits of this association are both cultivar- and fungus-dependent. Our objective was to assess the effect of inoculation with three native fungal strains (*Oidiodendron maius* A, *O. maius* BP, and *Acanthomyces lecanii* BC) on plantlet growth, plantlet survival, and nitrogen (N) absorption of the southern highbush blueberry (SHB) cultivars Biloxi and Misty. The fungal strains were inoculated into the peat-based substrate for growing blueberry cultivars, and plantlets produced by micropropagation were transplanted and grown for four months. The three inoculated strains positively affected the survival percentage in at least one of the cultivars tested, whereas *O. maius* BP positively affected plant biomass, N derived from fertilizer absorption, N content, and plant N recovery (%) in both Biloxi and Misty. Our results show that the *O. maius* BP strain may prove useful as a bio-inoculant to improve blueberry production during the nursery stage.

## 1. Introduction

Commercial blueberries can be divided into four groups, being the newest one the southern highbush blueberry (SHB). Plants of this group have larger fruit sizes and higher yields than the older blueberry groups but are the same as plants of the other groups, characterized by an inefficient root system with few root hairs. This feature can be compensated by their association with mycorrhizal fungi since fungal hyphae enhance the root’s ability to capture water and nutrients [1,2,3]. These fungi can also change the host root architecture and stimulate its colonization by modifying the concentrations of cytokinins, auxins, and strigolactones [4,5,6]. In this regard, Villarreal-Ruiz et al. [7] found that the inoculation of SHB with mycorrhizal fungi such as *Oidiodendron griseum* Robak and *Hyaloscypha hepaticicola* (D.J. Read) Korf and Kernan promotes an increase in root biomass and stimulates nutrient absorption, although the results were related by the fungus and/or nutrient used. Moreover, almost half a century ago, Pearson and Read [8] and Stribley and Read [9] found that, unlike non-inoculated plants, cranberry (*Vaccinium macrocarpon* Ait) mycorrhized plants showed increases in yield and nitrogen (N) concentration and were able to absorb N from organic residues. Non-mycorrhizal endophytic fungi, such as *Phialocephala fortinii* or *Acanthomyces* spp., can also stimulate plant host growth and nutrient uptake [1,10,11].

In blueberries, the commonly associated *Oidiodendron* sp. stimulates N absorption [12,13]. Nitrogen constitutes the most important nutrient to promote the vegetative growth of blueberries during the commercial propagation process [14]. In these juvenile plants, unlike that observed in most fruit crops, the absorption of N is more efficient in ammonia form [15,16]. Because of this, blueberry plants are fertilized with ammonium sulfate, which provides ammonia for plant nutrition, while the acidity generated contributes to keeping low values of soil pH [17], which generally favors the growth of soil fungi. In adult blueberry plantations, fertilization with ammonia also increases fruit production [18,19,20,21]. Fertilization supplies nutrients to meet plant needs. However, N applied to grown plants often moves with applied water and leaches and then can result in contamination of ground and surface water. Worsening this fact, substrates in containers are often leached to prevent substrate salinization. That is why the container plant industry, such as the blueberry plantlets production, is potentially an important source of environmental contamination [22].

Blueberry plants are obtained by micropropagation. One of the benefits of micropropagation is that it allows obtaining plants under conditions of total asepsis [23], which is an ideal condition for fungal inoculation. On the other hand, through micropropagation, plants are of longer shoots and have a larger root system than plants propagated by cuttings [23]. After micropropagation, plantlets are transplanted to a substrate suitable for their requirements. 

Powell and Bagyaraj [24] and Scagel et al. [25] found that, during SHB plant propagation, inoculation with mycorrhizal fungi positively affects growth and biomass, whereas Noé et al. [26] found that inoculation with mycorrhizae not only favors the growth of SHB but also decreases the use of fertilizers in the production process. Yang et al. [27] found that mycorrhized blueberry plants show higher biomass than non-mycorrhized ones because plants absorb N from soil organic residues through the action of mycorrhizae. This effect has been observed both in plants in production and in the in vitro propagation stage [28,29]. Meanwhile, Scagel [30] observed that the effects on N absorption in blueberry inoculated with mycorrhizal fungi were dependent on the cultivar, fungus inoculated, and fertilizer used. However, few reports have shown that the association of blueberries with fungi does not alter N absorption compared with non-inoculated plants [31]. 

Based on the above-mentioned facts and considering the need to improve N absorption from fertilizer and the effects of microbial activity, we developed the objective of the present study. We assess the influence of inoculation of native endophyte fungi isolated from two blueberry Argentine production areas on N absorption from fertilizer and growth of plantlets of two SHB cultivars of different vigor and phenological cycle. The fungi selected were two strains (BP and A) of the ericoid mycorrhizal fungus *Oidiodendron maius* and one strain (BP) of the endophytic fungus *Acanthomyces lecanii.*

## 2. Results

Root colonization was confirmed in all inoculated treatments. For *O. maius* strains, microsclerotia were observed in the cortical parenchyma of the root, and hyphae were observed entering through the epidermis of the root cells (Figure 1). 

Table 1 shows the results of total biomass (aerial + root) production, plantlet height, and survival percentage measured in both cultivars. Biomass showed no significant differences for the fungus × cultivar interaction or between cultivars but did show significant differences between the plantlets treated with the different fungi and the control. Plantlets associated with *O. maius* BP (T3) presented higher biomass than those associated with the other fungi or the control (*p* = 0.0244) (Table 1). Plantlet height showed no fungus × cultivar interaction and showed no differences among treatments, although T3 plantlets presented almost twice the height of control plantlets, probably due to data variability. Survival percentage showed fungus × cultivar interaction (*p* = 0.001). Plantlets of the cultivar Biloxi in the treatments T3 and T4 presented higher survival percentages than those in the treatments T1 and T2 (*p* = 0.0011), whereas plantlets of the cultivar Misty in the T2 and T4 treatment had higher survival percentages than those from T1 and T3 treatments (*p* ≤ 0.05) (Table 1).

Since there are no significant differences between cultivars, the averages of the total N, N content, N derived from the fertilizer, and % of N recovered are shown in Table 2. 

Plantlet N concentration showed significant differences among treatments, with T2 presenting the lowest value and the control and T4 treatments presenting the highest values. The N content (biomass × N concentration) found in the plantlets showed significant differences among treatments (*p* ≤ 0.05), with T3 showing the highest values.

The NdfF showed statistical differences among treatments only at 10% (*p* = 0.07), where plantlets from the T3 treatment absorbed twice as much N as those from T2 and T4 and five times more N than those grown in the control (T1). Although plantlets from T2 and T4 presented 60% and 63% more NdfF than those from T1, they showed no statistical differences from those from T1 (Table 2). The average data of aerial biomass analysis were: P 0.16%, K 0.99%, Ca 0.23%, Mg 0.19%, S 0.15%, B 107 mg kg^−1^, Cu 260 mg kg^−1^, Fe 159 mg kg^−1^, Mn 285 mg kg^−1^, and Zn 65 mg kg^−1^. The average data of root biomass analysis were: P 0.08%, K 0.52%, Ca 0.13%, Mg 0.10%, S 0.13%, B 101 mg kg^−1^, Cu 183 mg kg^−1^, Fe 416 mg kg^−1^, Mn 385 mg kg^−1^, and Zn 62 mg kg^−1^.

## 3. Discussion

Several studies reported the positive effects of mycorrhizal inoculation on the increase in plant biomass in blueberries, although not in all the varieties tested or in all the organs analyzed [2,25,30,32]. In our study, the positive effect of fungal inoculation on the plant biomass production of the two SHB cultivars studied was only verified in the treatment inoculated with the ericoid mycorrhizal fungus *O. maius* BP. According to Scagel [30], biomass production and nutrient absorption in blueberries are dependent on the inoculated fungus, indicating that plant–fungi interactions are dependent on the genotype of both partners. Relatively similar results have been found by Wazny et al. [11] and Grelet et al. [33]. The plant survival varied with the inoculated fungus, even between the treatments inoculated with both *O. maius* strains: A improved plant survival in both SHB cultivars whereas BP only in Biloxi.

In concordance with our results, Grelet et al. [33] also found differences in plant survival associated with different ericoid mycorrhizal fungi, even between two closely related strains. This was probably mediated by strain-dependent fungal effects on root growth or rates of root colonization, as was observed in several studies [11,12,13,14,15,16,17,18,19,20,21,22,23,24,25,26,27,28,29,30,31,32,33,34]. 

Regarding nitrogen absorption from fertilizer, *O. maius* BP presented the highest efficiency in supplying NdfF to the plants, which was reflected in the absorption and content of N. Similarly, Wei et al. [13] showed that inoculation with *O. maius* Om19 strain promotes N absorption and the growth *of Rhododendron fortunei,* and the genes related to N uptake and metabolism were highly upregulated in plants inoculated with mycorrhizal fungi.

The concentration of N showed some negative relation with plant biomass, which can be associated with the known “dilution effect” of the nutrients. Normally, the organs that reach the highest biomass have a lower percentage of N [35,36]. In blueberry plants, Throop and Hanson [37] and Bañados et al. [38] found a very low percentage of ^15^N from fertilizers (1%). However, Bañados et al. [38] also found that ^15^N absorption increased to 12–17% two months after application. Considering that our experiment lasted four months and that we harvested the plantlets three months after being fertilized with ^15^N, the value obtained in non-inoculated plantlets is within the same range as that found by Bañados and Strick [39]. However, the recovery of the N applied in plantlets inoculated with *O. maius* BP three months after fertilization was extremely high since it coincides with the results reached by adult blueberry plants after six months of applying the fertilizer [39,40]. This high N recovery is remarkable because, in order to absorb the N from the fertilizer, our plantlets had to first develop their root system. Inoculation with symbiotic mycorrhizal fungi can improve the rooting and root growth of blueberry plantlets by multiple mechanisms such as the production of auxins, which play an important role in the association of mycorrhizal fungi with plant roots [41,42]. Wu et al. [43] found that *Anteaglonium*, a DSE fungus associated with wild blueberry, had a growth-promoting effect on the host since its presence modulates the synthesis of phytohormones and flavonoids. Auxins have a direct effect on root architecture and development and increase root branching [44,45].

The nutrient concentrations in plant tissues are within the leaf normal range. With the exception of Ca, which was within the limit of deficiency, most nutrients are near the upper limit of normality from a current standard [46]. Except for Fe and Mn, nutrient concentration was lower in roots, as usual. 

Regardless of SBH cultivars, the two strains of mycorrhizal fungus *O. maius* showed positive inoculation effects on plantlets. Each strain affected the plantlets in different ways: one improved plant biomass and N absorption, whereas the other one increased the plant survival percentage. The use of *O. maius* BP as a bioinoculant would allow for the improvement of nitrogen use efficiency from fertilizers and to reduce nutrient leaching in blueberry production during the nursery stage. 

## 4. Materials and Methods

The experiments were carried out with SHB plantlets, obtained by micropropagation, growing in a culture room with controlled light and temperature (12 h of daily light and 20 °C average) since their implantation. The cultivars used were Misty, which has low to medium vigor [47], and Biloxi, which has high vigor [48]. Both cultivars are of early ripening fruit.

The fungal strains *Oidiodendron maius* BP, *O. maius* A, and *Acanthomyces lecanii* BC were utilized for inoculation experiments. The three strains were originally isolated from the roots of SHB plants growing in different blueberry production areas of Argentina and were identified using the internal transcribed spacer (ITS) region of rRNA sequence analysis [49]. The two *O. maius* strains selected present differences in their colony morphology, as shown in Figure 2.

For inoculum preparation, each strain was grown on malt extract agar in Petri dishes at 25 °C in the dark for 15 days. Then, 5 mm diameter discs were taken from the active growing zone of the fungal colonies and were single inoculated in a sterile substrate containing 32% organic matter and pH 4.5 maintained between 60% and 80% of field capacity and incubated at room temperature (25 °C) for 20 days. The substrate was integrated by a mix of a local equivalent of peat (based on river waste):perlite:sand at a 4:1:1 ratio based on volume. Plastic containers (8 cm W × 10 cm L × 7 cm H) were filled with 300 cm^3^ of the substrate, and eight micropropagated plantlets with at least four internodes of each SBH cultivar were transplanted. Labeled ammonium sulfate ((3% labeled (^15^N of 98+ atom% Aldrich) and 97% commercial ammonium sulfate) was applied in the four treatments further described one month after plantlets were planted. At that moment, plantlets began to grow, an indication that the root system was developing. Five successive fertilizer applications were made, one or two weeks apart, independently of treatments, to meet plantlets requirements as the plantlets grow. The total N added was 0.0094 gN/pot. The experiment lasted four months.

The experimental design was a completely randomized block design with four replications and a factorial arrangement. For each SHB cultivar, a total of 4 treatments were applied: (T1) substrate without fungal inoculum; (T2) substrate + *A. lecanii* BC; (T3) substrate + *O. maius* BP; and (T4) substrate + *O. maius* A. 

Plantlets were extracted from the substrates after four months of plantlets were transplanted, and the number of surviving plantlets and their height were recorded. Some of them were reserved to confirm the success of colonization of each fungus used. For this purpose, Trypan blue staining was performed according to the methodology proposed by Phillips and Hayman [50]. Plantlets were dried in an oven until constant weight (60 °C), and dry weight was recorded. Dry total biomass plantlets were ground, and total N and ^15^N concentrations were determined in the MBL Stable Isotope Laboratory (MA, USA) with a mass spectrophotometer (Europa 20-20 with a Europa ANCA-SL). The N derived from fertilizer (NdfF) was calculated using the following formula:NdfF = atom % ^15^N of the sample − atom % ^15^N natural abundance/atom % ^15^N of fertilizer − atom % ^15^N natural abundance

Phosphorus (P), potassium (K), calcium (Ca), magnesium (Mg), sulfur (S), boron (B), copper (Cu), iron (Fe), manganese (Mn), and zinc (Zn) were determined in the aerial and root biomass, using standard methodologies [51].

Results were subjected to statistical analysis using INFOSTAT [52]. The effect of the treatments on the variables and/or the interaction between the fungus × cultivar was analyzed by LSD Fisher at 5% significance. Since this interaction was significant for the survival percentage variable, each cultivar was analyzed independently using SAS [53].

## 5. Conclusions

The blueberry cultivars studied showed some similarities but different responses according to the fungal strain inoculated. *A. lecanii* did not represent a beneficial fungus, whereas *O. maius* affected the plantlets, although its effect depended on the strain inoculated. *O. maius* BP positively affected the absorption of ^15^N and, therefore, the value of NdfF and the N recovery percentage, thus increasing the N content and the biomass of the plants of both varieties. *O. maius* A improved plant survival, helping in plant establishment. The positive effects of the *O. maius* BP strain showed that it has a high potential for application in the management of blueberry nutrition during the nursing stage. In a broader vision, not only the endophyte fungi species are important for the behavior of blueberry plantlets, but also its strains. That is why the best strains must be tried to find to achieve the goal of producing blueberry seedlings in a more environmentally friendly way.

Possibly due to the short time for plantlets production, no major differences between blueberry cultivars were found.

## Figures and Tables

**Figure 1 plants-12-00792-f001:**
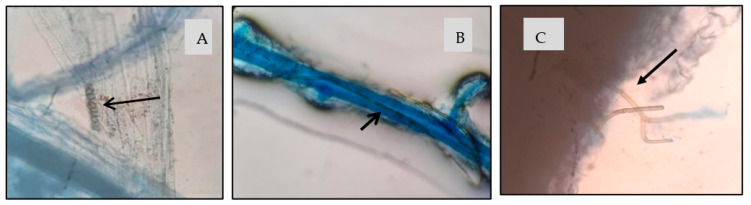
SHB roots inoculated with *O. maius* (**A**) microsclerotia (pointed by the arrow) in cortical parenchyma; (**B**,**C**) hyphae entering the root cells.

**Figure 2 plants-12-00792-f002:**
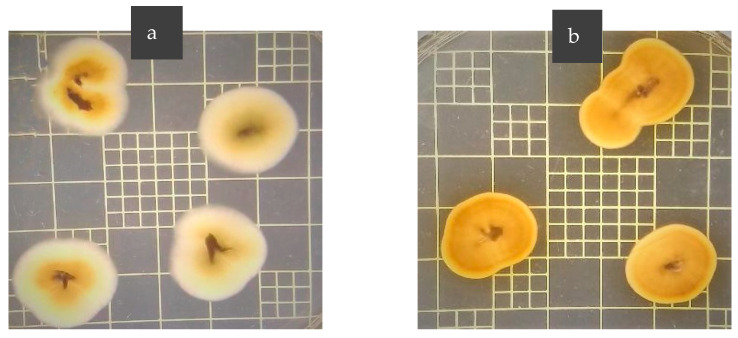
*Oidiodendron maius* BP (**a**) and *O. maius* A (**b**) colonies grown for two weeks in malt agar.

**Table 1 plants-12-00792-t001:** Fungal inoculation effect on total plant biomass (dry mass), plant height, and survival of Biloxi and Misty southern highbush blueberry cultivars.

Inoculation Treatment	Total Biomass (mg)	Plant Height (cm)	Survival (%)	
			Biloxi	Misty
T1 (Non-inoculated)	8 b	2.55 a	4.68 b	17.18 b
T2 (*A. lecanii* BC)	12 b	3.07 a	4.68 b	34.37 a
T3 (*O. maius* BP)	20 a	4.80 a	43.75 a	14.06 b
T4 (*O. maius* A)	11 b	4.20 a	43.75 a	43.75 a
Significance	*	NS	*	*
**Cultivar**				
Misty	11	3.55	27.34	
Biloxi	13	3.79	24.22	
Significance	NS	NS	NS	
Fungus × Cultivar	NS	NS	*	

Mean values are shown. For each variable, different letters show significant differences between treatments (*p* < 0.05). NS: not significant, * *p* < 0.05.

**Table 2 plants-12-00792-t002:** Fungal inoculation effect on N content, nitrogen derived from the fertilizer (NdfF), and N recovered percentage found in both southern highbush blueberry cultivars (Misty and Biloxi plantlets).

Inoculation Treatment	Total Nitrogen (%)	Nitrogen Content (N mg/plantlets)	NdfF	Plant ^15^N Recovery (%)
T1 (Non-inoculated)	3.45 a	0.124 b	0.04 b	10 *
T2 (*A. lecanii* BC)	2.44 c	0.069 b	0.09 b	5 *
T3 (*O. maius* BP)	2.98 b	0.265 a	0.20 a	26 *
T4 (*O. maius* A)	3.12 ab	0.169 ab	0.10 b	11 *
Significance	**	**	*	NS

Mean values are shown. For each variable, different letters show significant differences between treatments (*p* < 0.05). NS: not significant, * *p* < 0.05, ** *p* < 0.01.

## Data Availability

All data are available upon request to María A. Pescie.

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
