# Peer review of "Inoculation with Oidiodendron maius BP Improves Nitrogen Absorption from Fertilizer and Growth of Vaccinium corymbosum during the Early Nursery Stage"

_plants, 2023, doi:10.3390/plants12040792_

Round 1

Reviewer 1 Report

The manuscript entitled “  “ is an interesting and actual thematic subject that fits within the area of this journal. Although very clean and synthetical presented this manuscript needs more clarification.

Introduction is short and concise.

Material and Methods section you said: “Labeled ammonium sulfate [(3% labeled (15N of 98+ atom% Aldrich) and 97% commercial ammonium sulfate] was applied one month after plantlets were planted. At the moment plantlets began to grow, an indication that the root system was developing. Five successive fertilizer applications were made, one or two 203 weeks apart, to meet plantlets requirements.” Was this independently of treatments you described later on? Please define the total amount of N added. 

Plant growth characteristics: type of substrate; pot volume; climate; full time of experiment.

The experimental design was a completely randomized blocks design with a total of 2x4x4= 24 plants ??

There is a missing control: no fungus + no fertilizer

“Dry plantlets were ground and total N ..” Means what ? Total plantlets? 

Why only aerial biomass was used for the determination of other macro and micro elements?

Results section is clear and concise. However there are missing aspects: data of aerial biomass analysis were similar in all treatments? Why N + NdfF + N recovery were not presented considering the 2 cultivars? Because the survival rate was too low for Biloxi?

Discussion section is too short. How do you explain the differences between the 2 cultivars in terms of survival? Is this due a huge variability of responses? It’s strange being plantlets that should be clones! On the other hand the differences - though short - between O.maius A and BP. Also, why did you use A. lecanii BC? 

The action of O. maius on plants may be associated with the production of auxins…yes, it’s true but why more in BP than in A?

Finally in conclusions you talk only about fungus inoculation but you tested also 2 plant cultivars. That presented different results. Also, what is your advise for other researchers? Choose or isolate the best fungi strains according to the region ? What?

Author Response

Attached is our response.

Reviewer 2 Report

María A. Pescie and the co-authors have written the paper entitled Inoculation with Oidiodendron maius BP improves nitrogen absorption from fertilizer and growth of Vaccinium corymbosum during the early nursery stage. The paper assesses the influence of inoculation of native endophyte fungi isolated from two blueberry Argentine growing areas on N absorption from fertilizer. The other aim was focused on the impact of the selected fungi strains on the growth of plantlets of two Southern Highbush Blueberry cultivars of different vigor and phenological cycle.

The authors have presented in the introduction the expected facts, e.g., that

 fungal hyphae enhance the root's ability to capture water and nutrients; fungi can also change the host root architecture and stimulate its colonization by modifying the concentrations of cytokinins and auxins; fungi promote an increase in root biomass and stimulating nutrient absorption; 

However, for a scientific paper, a broader context is necessary. It has to be remembered that agriculture is not producing. Agriculture is able only to enhance the natural biological processes. Currently, agriculture is causing much environmental degradation (wetland drainage, using harmful amounts of herbicides and pesticides) because the natural biogeochemical basics of agriculture are ignored. The application of plant -the fungi relationship is an example of applying biological processes into agriculture practices.

The scientific paper must present the biological background of the studied processes and identify and describe processes. The plant fungi symbiosis is possible only if the traits of both involved organisms allow the cooperation to run.

Another area for improvement of this paper is the need for a deep discussion of the extent to which the strictly controlled experimental conditions are controlled. It is well known that differences in habitat conditions cause the modification of the character of the organism's mutual relation such as, e.g., stimulation of N absorption and increase in below and above-ground biomass.

I would appreciate it if the author put the corrections in red to be better indicated in the text.

Author Response

Attached is our response.

Round 2

Reviewer 1 Report

You need to correct the last sentence of conclusions. "Life" is missing.

Reviewer 2 Report

The authors improved the paper significantly. I am happy with the corrections.